# Characterization of Wingbeat Frequency of Different Taxa of Migratory Insects in Northeast Asia

**DOI:** 10.3390/insects13060520

**Published:** 2022-06-03

**Authors:** Wenhua Yu, Haowen Zhang, Ruibin Xu, Yishu Sun, Kongming Wu

**Affiliations:** 1Institute of Insect Sciences, College of Agriculture and Biotechnology, Zhejiang University, Hangzhou 310058, China; wenhua.yu@zju.edu.cn; 2Key Laboratory for Biology of Plant Diseases and Insect Pests, Institute of Plant Protection, Chinese Academy of Agricultural Sciences, Beijing 100193, China; marszhang_0@163.com (H.Z.); xuruibin413@126.com (R.X.); sunyishuwy@163.com (Y.S.); 3State Key Laboratory of Ecological Pest Control for Fujian and Taiwan Crops, Institute of Applied Ecology, Fujian Agriculture and Forestry University, Fuzhou 350002, China; 4Agricultural Information Institute, Chinese Academy of Agricultural Sciences, Beijing 100081, China

**Keywords:** insects, migration, wingbeat frequency, morphometrics, Northeast Asia

## Abstract

**Simple Summary:**

Wingbeat frequency (WBF), an important variable in the study of flight biology, is very valuable in identifying migratory behavior. Thus, the WBF of migratory insects in Northeast Asia was detected and analyzed to establish the relationship between WBF and insect morphometrics. The results demonstrated that WBF differed across orders and that morphological variables were closely connected to this observed variation. This study may be helpful for increasing our understanding of flight biology and for developing new methods to identify the species of migrating insects.

**Abstract:**

The ability to migrate is an important biological trait of insects, and wingbeat frequency (WBF) is a key factor influencing migratory behavior. The WBF of insects has been shown to be species-specific in previous studies; however, there is scant information on variations in WBF among different taxa of migratory insects. In 2018 and 2019, we investigated the relationship between WBF and 12 morphological variables (e.g., body mass, body length, total wing area, etc.) of the main migratory insects (77 species in 3 orders and 14 families) over the Bohai Sea in China. The WBF of migratory insects was negatively correlated with the 12 morphological variables and varied significantly among orders. In migratory lepidopterans, neuropterans, and odonatans, the ranges of WBF were 6.71–81.28 Hz, 19.17–30.53 Hz, and 18.35–38.01 Hz, respectively. Regression models between WBF and connecting morphological variables were established for these three orders. Our findings revealed the relationship between WBF and morphometrics of migratory insects in Northeast Asia, increased our knowledge on the flight biology of migratory insects, and provided a basis for developing morphological and WBF-based monitoring techniques to identify migrating insects.

## 1. Introduction

Insect migration profoundly impacts ecosystems by contributing to seasonal flows of energy, nutrients, and pathogens. In response to adverse environments, insects migrate on a large scale, which also affects human activities [1,2]. Migratory insects may be considered pests or beneficial. Most insect pests such as the fall armyworm, *Spodoptera frugiperda* (J.E. Smith), are lepidopterans, and their long-distance migration en masse to different agricultural regions can lead to significant crop losses [3,4,5,6]. However, natural enemies of some lepidopterans, such as dragonflies, can suppress pest populations [7]. However, the decrease in insect biodiversity due to environmental factors such as climate change [8,9,10] has led to declines in beneficial insects. For example, the migratory dragonflies, which are the natural enemies of pests, have declined at a rate of 14.1% per year [11]. This will have a negative impact on the stability of the ecosystem balance. Monitoring different taxa of migratory insects is important for understanding and predicting fluctuations in pest and beneficial insect populations.

The characteristics of wings and wingbeats are critical for insects to take off and complete migration. For example, the amplitude of wing vibrations gradually increases during the first 10–14 wingbeats of droneflies so that they can take off after about 12 beats [12]. When aphids stop beating their wings, they can fall to the ground from an altitude of 1000 m in 20 min [13]. Wing form influences the magnitude of the force; for example, the combined action of the front and rear wings of dragonflies and lepidopterans promotes force production, while the front and rear wings of lepidopterans are connected, making the wing bases wider [14]. Different migratory insects have evolved different external morphological characteristics suitable for long-distance migration using air currents. For example, the black cutworm (*Agrotis ypsilon*), a lepidopteran pest, has broader outer wings. In addition, migratory dragonflies have longer wing lengths than nonmigratory insects in the same family [15]. Similarly, males of the dark-veined pink butterfly *Pieris napi* have greater weight loads and slower flight speeds than females [16].

Wingbeat frequency (WBF) is also a critical variable for insect flight activity and is influenced not only by the physiological state of the insect and air temperature [17] but also by the size of the insects. During wing vibration, smaller bodied insects typically have higher WBF than larger ones to overcome aerial drag and to create sufficient lift [18]. However, it is not clear how the size of different migratory insects affects WBF. In addition, WBF is an important biometric variable for identifying and monitoring migratory insect species [19].

In the present study on Beihuang Island in the Bohai Strait, which lies on an important migration route for insects in Northeast Asia, we measured the WBFs of lepidopterans and dragonflies (odonatans) [11], as well as neuropterans (e.g., lacewings and important natural enemies [20]), which are commonly trapped on the island during their migration. We tested for the correlation between WBF and 12 external morphological variables (e.g., body mass (BM) and total wing area (TWA)). Our study showed WBF differences among insect orders, families, and species. WBFs were correlated with morphological variables. Regression models between WBF and morphometrics were built. This study will improve our knowledge of the flight biology of migratory insects and provide a basis for the development of morphological and WBF-based techniques to identify and monitor migratory insect populations.

## 2. Materials and Methods

### 2.1. Study Specimens

During May to October in 2018 and 2019, migratory insects were caught using three searchlight traps on Beihuang (BH) Island (38°24’ N, 120°55’ E) in Shandong Province of China. This 2.5 km^2^ island is located in the middle of the Bohai Strait, 40–60 km away from mainland China. Each searchlight trap consisted of a 1000 W metal halide lamp (JLZ1000BT, Shanghai Yaming-Lighting Co Ltd., Shanghai, China), a lamp holder, and a 60-meshnylon bag [11] (Appendix A). Except during power outages caused by extreme weather, trapping was conducted every day. The migratory insects were collected from the searchlight traps in the morning, and the individuals who were both intact as well as freely moving were used for experiment. A total of 1458 specimens across 77 species in 14 families of 3 orders were tested in this study.

### 2.2. WBF Measurements

The insects for testing were first transferred to a dark climate incubator (RXZ, Ningbo Southeast Instruments, China) at a temperature of 25 ± 1 °C and a relative humidity of 75% ± 5% for acclimation for at least 10 min. Individual insects were then placed on a spreading plate to spread their wings, and 502 glue was dabbed onto the thoraco-ventral junction or thoracic dorsal plate of each insect, where a 0.25–1.0 mm diameter copper wire was attached (Ergo, Kisling AG, Wetzikon, Switzerland). This wire was then inserted into a Styrofoam panel, with the individual insect hanging below the panel. A stroboscope (Phaser-Strobe pbx, Monarch, Amherst, NH, USA) was subsequently used to measure and record the WBF of individual insects. Using the stroboscope, the frequency of flashing light was adjusted and the data were read from the screen of the apparatus when the insect’s wings remained visually static (Video S1) [21].

### 2.3. Morphological Measurements

After their WBFs were recorded, the test insects were frozen in a −20 °C freezer, and then, 12 morphological variables of each insect were measured (Table 1). Firstly, the complete body was weighed on a precision balance scale (CP224C, Ohaus, Shanghai, China), and the data shown on the screen of the apparatus were recorded as BM. We then cut off the wings using scissors and then measured the length and width of the insect were with an electronic digital caliper (7D-02150, Forgestar, Shanghai, China). The longest part of the body was recorded as body length (BL), while the widest part of the body was recorded as body width (BW) (Figure 1). After recording the BL and BW, we then cut off the thorax and abdomen of the insect with scissors, and weighed and recorded the thorax (without appendages) and abdomen separately with the balance scale. Later, forewing length (FWL), forewing width (FWW), hindwing length (HWL), hindwing width (HWW), forewing area (FWA), hindwing area (HWA), and TWA were measured by a super-field 3D microscope (VHX-2000, Keyence Corporation, Shanghai, China). During the measuring process, the actual length, width, and area of wings were obtained using scales and pixels. The longest and widest parts and the area of each wing were separately measured as the length, width, and area of the wings (Figure 1) [22]. The average length, width, and area of the left and right forewings were denoted as FWL, FWW, and FWA, respectively. The average length, width, and area of the left and right hindwings were denoted as HWL, HWW, and HWA, respectively. The total area of the two pairs of wings was recorded as TWA. After measuring those 12 single morphological variables, 7 connecting morphological variables (Table 2) were calculated.

### 2.4. Data Analysis

Data were first checked for normality and homoscedasticity. If data were normally distributed or demonstrated homogeneity of variance, analysis of variance was used; otherwise, a nonparametric test was used. The Kruskal–Wallis test was used to analyze inter- and intra-order variation in WBF. Spearman’s correlation analysis was used to test for a correlation between WBF and morphological variables for different groups of insects. For taxa where WBF and morphological variables were correlated, a one-dimensional linear regression or nonlinear regression was used to analyze the relationships between WBF and morphological variables. The above analyses were performed using R language (version 4.4.1; R Core Team, 2021) [23].

A random forest classification model was utilized to classify insects into families based on WBF and morphological variables. The sklearn package’s GridSearchCV function was used to grid search the hyperparameters in order to achieve accurate classification with the best hyperparameter combination. To avoid overfitting, the original data were randomly divided into 10 duplicates, with 8 copies serving as the training set and the remaining 2 serving as the test set. The cross-validation was conducted 10 times, and the average of the 10 accuracies was taken as the evaluation index of the final model. The random forest model’s parameter space was set to the number of trees in the forest [100, 1000] with steps of 50, and the random forest’s depth was set to [10,20] with steps of 1. Analyses were conducted using Python (3.7.3).

## 3. Results

The WBFs of the insects measured ranged from 6.71 to 81.28 Hz (Appendix A), with significant differences among the three orders of insects. Lepidopterans had significantly higher WBFs than odonatans, which had significantly higher WBFs than neuropterans (Figure 2a). WBFs of lepidopterans ranged from 6.71 to 81.28 Hz, with significant interspecific differences; saturniids had the lowest average WBF (Figure 2b) [21]. WBFs of neuropterans ranged from 19.17 to 30.53 Hz; *Chrysoperla sinica* (Tjeder) had the highest mean WBF (Figure 2c). WBFs of dragonflies ranged from 18.35 to 38.01 Hz; no significant interspecific differences in WBF were found (Figure 2d).

There were significant negative correlations between WBF and the 12 morphological characters (i.e., BM, thorax mass (TM), abdominal mass (AM), BL, BW, FWL, FWW, FWA, FWL, HWW, HWA, and TWA) for all test insects except for the odonates. Connecting variables such as the square root of body mass divided by total wing area (BM^1/2^/TWA) or the square root of body mass divided by the product of wing length and wing width (BM^1/2^/(FWL × FWW)) have large influences on the WBF of either all test insects or only lepidopterans (Figure 3).

Regression analysis of WBF against the 12 morphological characters for all test insects showed a significant negative correlation (*p* < 0.001) between WBF and each characteristic (i.e., BM, TM, AM, BL, and BW), but the correlation was weak (*r*^2^ < 0.1). There was an exponentially decreasing correlation (0.26 < *r*^2^ < 0.45) between WBF and wing characteristics (i.e., HWW, TWA, FWA, HWA, FWW, HWL, and FWL) (Figure 4). The regression analysis of WBF and seven connecting morphometrics for all test insects showed that the BM^1/2^/TWA was the best predictor of WBF (*r*^2^ = 0.65). WBF increased as the variable BM^1/2^/TWA increased (Figure 5).

The regression analysis of WBF and 12 morphological characters of only the lepidopterans showed a significant trinomial or binomial regression relationship of WBF with body morphometrics (i.e., BM, TM, AM, and BL) (*r*^2^ < 0.1, *p <* 0.05). There was an exponentially decreasing correlation (0.36 < *r*^2^ < 0.60, *p <* 0.001) between WBF and each wing morphometric (i.e., HWW, TWA, FWA, HWA, FWW, HWL, and FWL) (Appendix A). A regression analysis of WBF and seven combinations of morphological characteristics for all test lepidopterans showed that the BM^1/2^/TWA was the best predictor of WBF (*r*^2^ = 0.67). WBF increased as the variable BM^1/2^/TWA increased (Figure 6).

For the neuropterans measured, there was a positive binomial regression relationship (*p <* 0.001) between WBF and 10 morphological characters (i.e., BM, TM, AM, FWL, FWW, FWA, HWA, HWW, HWA, and TWA). The influences of FWL, TWA, FWA, and HWA on WBF were remarkable (*r*^2^ ≥ 0.50). Although WBF tended to decrease and then increase in relation to each wing morphometric, overall WBF decreased with increasing wing measurements (Appendix A). The results of the regression analysis of WBF and the seven combinations of morphological morphometrics for all test neuropterans showed that the combinations of morphological characters were poor predictors of WBF (*r*^2^ < 0.12) (Figure 7).

The classification accuracy was 92.07% when 13 biometric variables (i.e., WBF, BM, TM, AM, BL, BW, FWL, FWW, FWA, HWL, HWW, HWA, and TWA) of families were used. Based on four biometric variables (WBF, BL, BW, and BM) that can be measured by the insect radar, the classification accuracy was larger when using all four variables (84.5%) rather than only using three, two, or just one variable (Table 3).

## 4. Discussion

This study demonstrated that the morphometrics (e.g., BM and TWA) inversely affected WBF, consistent with the results of Tercel et al., who found a clear tendency for lower WBFs in insects with greater BM or wing size (especially wing area) [24]. Other authors reported similar results for *B. ignitus* [22] and *Aedes* [*Stegomyia*] *aegypti* [25]. Notably, wing morphometrics were more closely related to WBF than body morphometrics were in our study. A previous study showed that BM has no effect on the WBF of some insects such as *B. terrestris* [22]. A larger wing means that the insect makes fewer vibrations in the same amount of time to balance the same amount of lift [18]. This study suggests that wing morphometrics has a large influence on WBF.

Our study revealed that BM^1/2^/TWA is a good predictor of WBF. This outcome was explained by Deakin, who used this equation to explain the positive correlation between WBF and BM^1/2^/TWA. [26]. BM and TWA did not increase or decrease proportionally; thus, the positive relationship further confirms that wing morphometrics are the main variables affecting WBF. In addition, a previous study showed that even though there is a weak correlation between WBF and BM, together with wing area, BM variables also contribute greatly to WBF [24]. Thus, WBF is influenced by BM and TWA to a large extent.

In the present study, BM^1/2^/TWA was a better predictor than other variables of WBF of lepidopterans. Lepidopteran insects have a relatively large wing base, and both the fore- and hindwings generate the forces required for insect flight [14]. This may explain why BM^1/2^/TWA predicts WBF better than BM^1/2^/FWA or BM^1/2^/HWA, but further study is needed on the contribution of these variables to WBF.

The WBF of neuropterans increased first and then decreased with increasing BM, FWL, or wing area in this study. The effect of body size on aerodynamics or metabolic rate of the insect during flight may explain this phenomenon. For some insects, the leading-edge vortex generated by the leading edge of the wing helps to generate lift as the wings beat [27]. The FWA and TWA have greater effects on WBF than the other variables. Thus, with longer wings, insects can beat fewer times per unit time to generate the same amount of lift. However, when insect size increases up to a certain size, the metabolic rate also needs to be considered, as metabolic rate during flight increases as body mass increases [28,29] and is positively correlated with WBF [30]. Therefore, the higher metabolic rate is likely to account for the higher WBF when the BM of a lacewing increases within a certain range.

The different wing movement mechanisms of insects may also influence the correlation between the WBF and morphometrics. In our study, the WBFs of dragonflies were not correlated with any morphological variable. This result may be related to the interacting forces between the fore- and hindwings of dragonflies, which help generate the forces required for flight by changing the phase of the front and rear wings [31]. The phase difference between the front and rear wings is sufficient for the fore- and hindwings to interact hydrodynamically [32] and to generate more lift [31]. However, when dragonflies beat their wings, the wings are tilted forward with a large angle of attack for the downbeat and a smaller angle of attack for the upbeat, with the forewing lagging behind the hindwing when the two wings interfere unfavorably. As the wings beat, lift and drag combine to support dragonfly flight, so it is possible that the effect of the fore- and hindwing interaction on WBF masks the effect of any size variable on WBF, which may explain why morphology does not have a significant effect on dragonfly WBF.

In the present study, we showed that there were significant inter- or intra-order differences in WBFs. WBF differences among insect orders and families could be important for the identification of insect species. Furthermore, the correlation between WBF and morphometrics (e.g., TWA) we established can be useful to estimate insect size by WBF or WBF by morphological variables. For example, with the lowest WBF in 11 families of Lepidoptera, Saturnnidae insects have the largest TWA (2964.08–5838.87 mm^2^) in this study. However, the difference in WBF between some insects was not significant, such as *Pantala flavescens* (18.85–38.01 Hz) and *Anax parthenope julius* (28.31–35.8 Hz) (Figure 2d). Thus, the identification of insects by WBF is not feasible, as insects with similar WBF may differ in size; for example, *P. flavescens* (weight 237.6–416.5 mg and length 46.27–52.28 mm) is smaller than *A. parthenope julius* (weight 697.4–1308.8 mg and length 66.54–73.75 mm). In addition, our attempts to classify insects into families based on 13 biometric variables had an accuracy of 92.07%. However, the WBF and morphological variables were correlated. WBF, BM, BL, and BW are important variables for radar discrimination of migrating insects [19,33]. When we used only four biometric variables to identify insects into families, our results were only 84.5% accurate. This study also shows that BM, BL, and BW do not have a strong relationship with WBF. These results will provide a theoretical basis for the automatic (radar) identification of insect species.

Flying is an energy-consuming activity [34,35]. Insect size can also affect energy conservation and utilization [36]. Our study showed that both WBF and the relationship between WBF and morphometrics varied among lepidopterans (i.e., mainly pests), neuropterans, and odonates (i.e., natural enemy insects). The WBF differences may indicate the potential differences in energy between migratory pests and natural enemies. Wing flexibility contributes to reducing the energetic requirements of insect flight [37]. The different morphological structures of different groups of insects may affect wing flexibility and, therefore, energetic requirements. Morphological variables (e.g., BM) linked between predator and prey influence energy flow [38]. Future studies will be necessary to focus on the relationships among body size, WBF, and energy metabolism to deepen our understanding of the mechanism between WBF and morphological variables.

## Figures and Tables

**Figure 1 insects-13-00520-f001:**
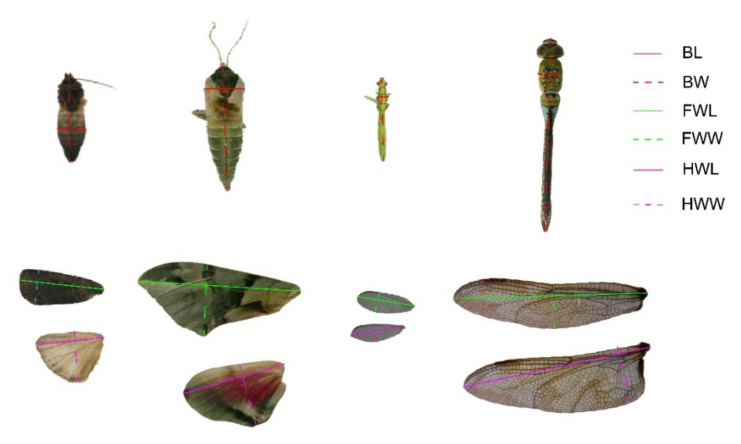
Schematic diagram of morphological variables related to length and width measured for insects. From left to right, each picture represented a lepidopterous insect, a lepidopterous insect, a neuropteran, and a dragonfly. BL indicates body length, BW indicates body width, FWL indicates length of forewing length, FWW indicates forewing width, HWL indicates hindwing length, and HWW represents hindwing width.

**Figure 2 insects-13-00520-f002:**
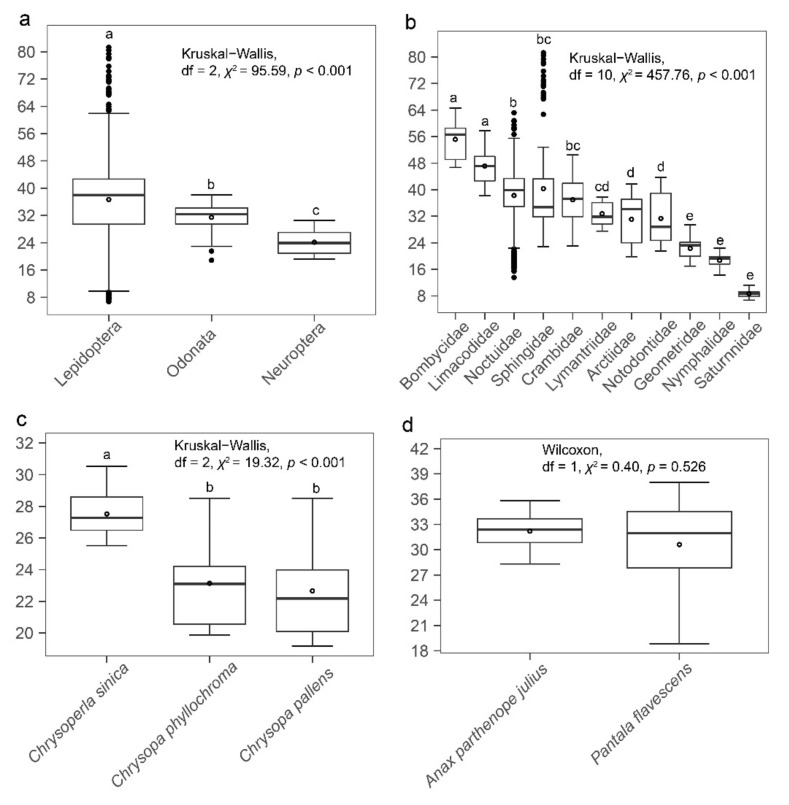
Differences in wingbeat frequency among (**a**) orders; (**b**) lepidopteran families, adapted from [21]; (**c**) neuropteran species; and (**d**) odonatan species. From bottom to top, each box-and-whisker plot represents the minimum, lower quartile, median, upper quartile, and the maximum, and the empty circle represented the mean. Bars with the same letter indicate no differences in WBF.

**Figure 3 insects-13-00520-f003:**
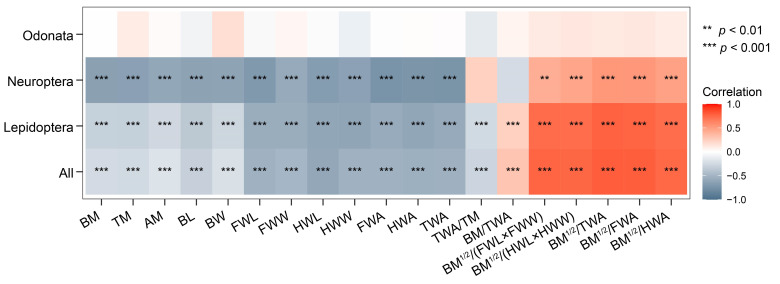
Correlations between wingbeat frequency and morphometrics. BM, body mass; BM^1/2^, the square root of body mass; TM, thoracic mass; AM, abdominal mass; BL, body length; BW, body width; FWL, forewing length; FWW, forewing width; HWL, hindwing length; HWW, hindwing width; FWA, forewing area; HWA, hindwing area; and TWA, total wing area.

**Figure 4 insects-13-00520-f004:**
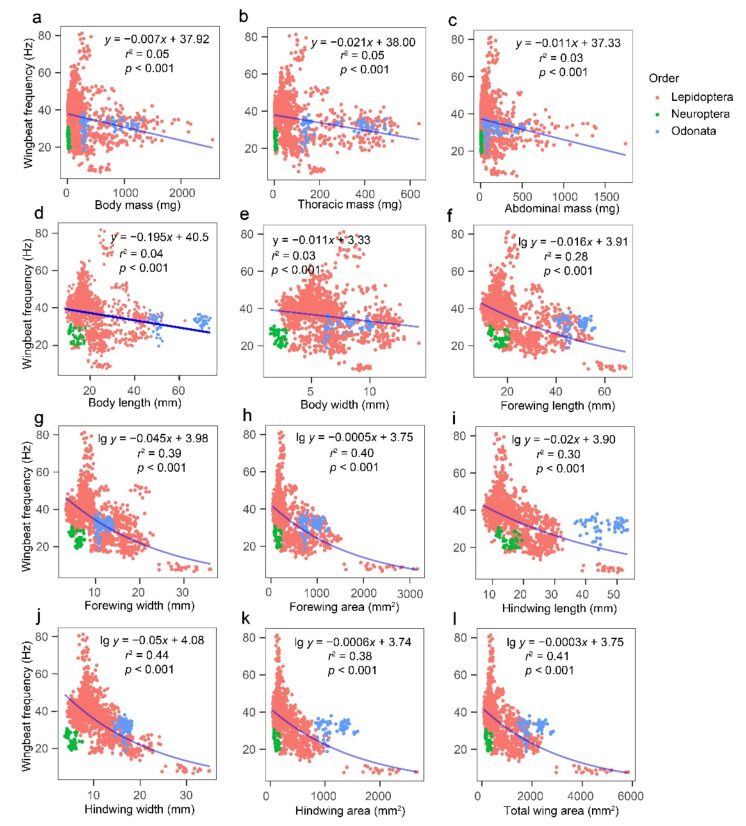
Relationships between wingbeat frequency and morphological variables (**a**) body mass, (**b**) thoracic mass, (**c**) abdominal mass, (**d**) body length, (**e**) body width, (**f**) forewing length, (**g**) forewing width, (**h**) forewing area, (**i**) hindwing length, (**j**) hindwing width, (**k**) hindwing area, and (**l**) total wing area, for all tested insects. *N* = 1458 individuals.

**Figure 5 insects-13-00520-f005:**
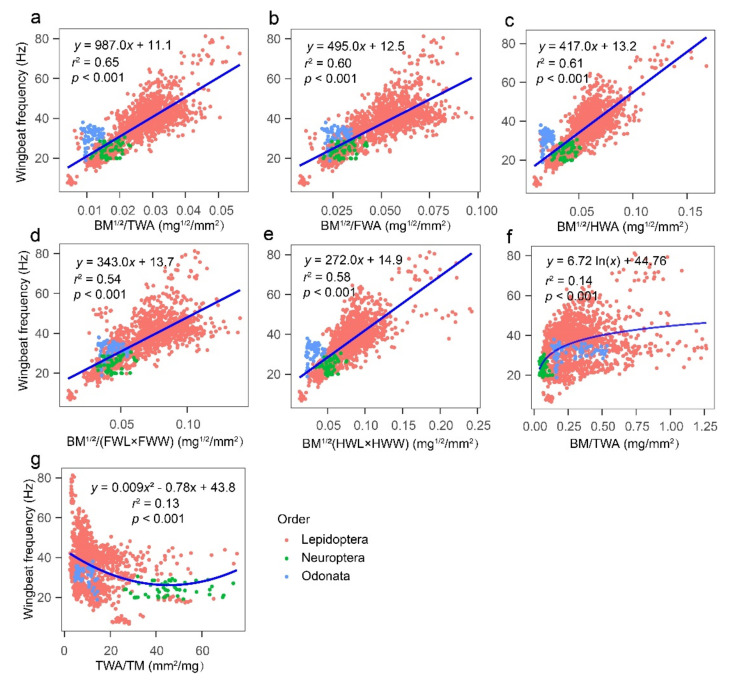
Regression models for wingbeat frequency and morphological variables (**a**) the square root of body mass divided total wing area, (**b**) the square root of body mass divided forewing area, (**c**) the square root of body mass divided hindwing area, (**d**) the square root of body mass divided by the product of forewing length and forewing width, (**e**) the square root of body mass divided the product of hindwing length and hindwing width, (**f**) body mass divided total wing area, and (**g**) total wing area divided by thoracic mass, for all tested insects. BM, body mass; TM, BM^1/2^, the square root of body mass; thoracic mass; AM, abdominal mass; BL, body length; BW, body width; FWL, forewing length; FWW, forewing width; HWL, hindwing length; HWW, hindwing width; FWA, forewing area; HWA, hindwing area; and TWA, total wing area. *N* = 1458 individuals.

**Figure 6 insects-13-00520-f006:**
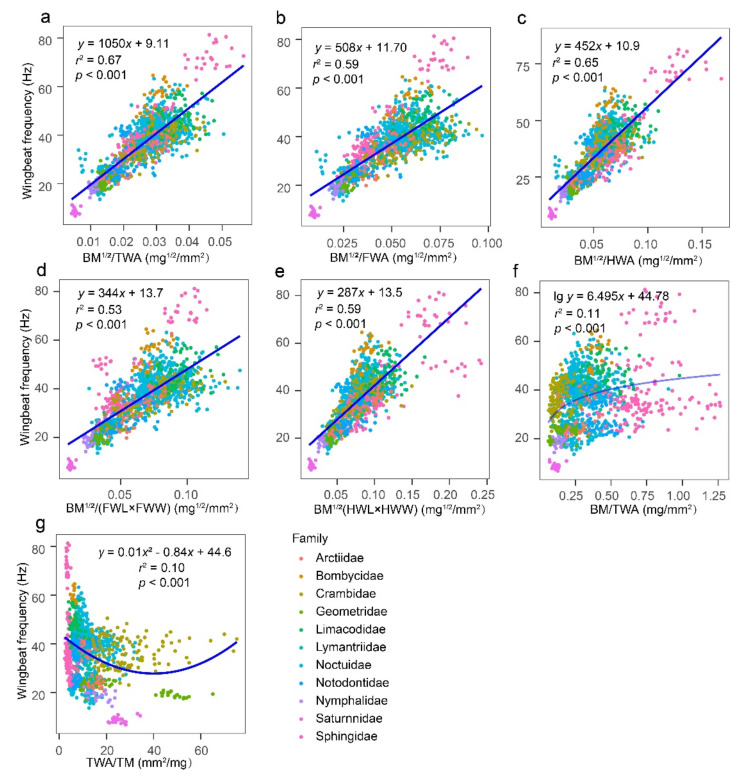
Regression models for wingbeat frequency and morphological variables (**a**) the square root of body mass divided total wing area, (**b**) the square root of body mass divided forewing area, (**c**) the square root of body mass divided hindwing area, (**d**) the square root of body mass divided by the product of forewing length and forewing width, (**e**) the square root of body mass divided the product of hindwing length and hindwing width, (**f**) body mass divided total wing area, and (**g**) total wing area divided by thoracic mass, for all tested lepidopterans. BM, body mass; BM^1/2^, the square root of body mass; TM, thoracic mass; AM, abdominal mass; BL, body length; BW, body width; FWL, forewing length; FWW, forewing width; HWL, hindwing length; HWW, hindwing width; FWA, forewing area; HWA, hindwing area; and TWA, total wing area. A total of 1344 individuals were used in this analysis.

**Figure 7 insects-13-00520-f007:**
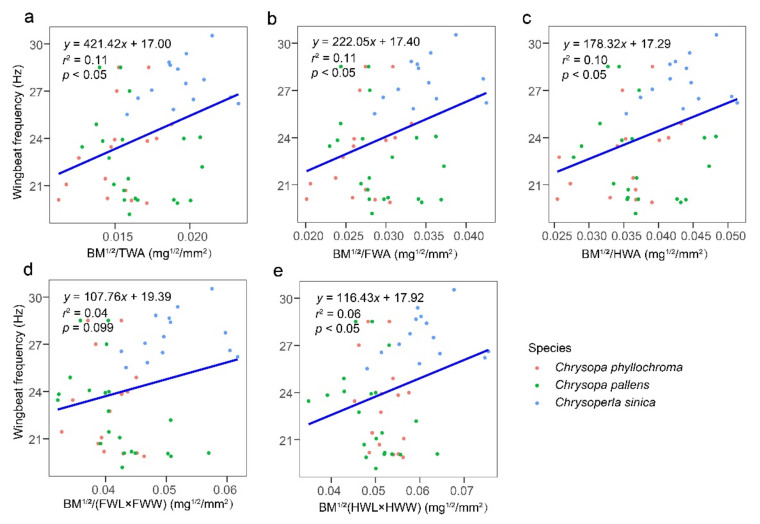
Regression models for wingbeat frequency and morphometrics (**a**) the square root of body mass divided total wing area, (**b**) the square root of body mass divided forewing area, (**c**) the square root of body mass divided hindwing area, (**d**) the square root of body mass divided by the product of forewing length and forewing width, and (**e**) the square root of body mass divided the product of hindwing length and hindwing width, of all test neuropterans. BM, body mass; TM, thoracic mass; AM, abdominal mass; BL, body length; BW, body width; FWL, forewing length; FWW, forewing width; HWL, hindwing length; HWW, hindwing width; FWA, forewing area; HWA, hindwing area; and TWA, total wing area. *N* = 51 individuals.

**Table 1 insects-13-00520-t001:** Twelve morphometrics of insect body parts measured in this experiment.

Morphological Variables	Abbreviation	Instruments	Meaning
Body mass	BM	A precision balance	Weight of body
Thoracic mass	TM	A precision balance	Weight of thorax
Abdominal mass	AM	A precision balance	Weight of abdomen
Body length	BL	An electronic digital caliper	The longest part of body
Body width	BW	An electronic digital caliper	The widest part of body
Forewing length	FWL	A super-field 3D microscope	The average of the left and the right forewing lengths
Forewing width	FWW	A super-field 3D microscope	The average of the left and the right forewing widths
Forewing area	FWA	A super-field 3D microscope	The average of the left and the right forewing areas
Hindwing length	HWL	A super-field 3D microscope	The average of the left and the right hindwing lengths
Hindwing width	HWW	A super-field 3D microscope	The average of the left and the right hindwing widths
Hindwing area	HWA	A super-field 3D microscope	The average of the left and the right hindwing areas
Total wing area	TWA	A super-field 3D microscope	The sum of the average areas for the two pairs of wings

**Table 2 insects-13-00520-t002:** Seven connecting morphological variables measured in this experiment.

Connecting Morphological Variables	Abbreviation
Total wing area divided by thoracic mass	TWA/TM
Body mass divided total wing area	BM/TWA
The square root of body mass divided by the product of forewing length and forewing width	BM^1/2^/(FWL × FWW)
The square root of body mass divided by the product of hindwing length and hindwing width	BM^1/2^/(HWL × HWW)
The square root of body mass divided by total wing area	BM^1/2^/TWA
The square root of body mass divided by forewing area	BM^1/2^/FWA
The square root of body mass divided by hindwing area	BM^1/2^/HWA

**Table 3 insects-13-00520-t003:** The accuracy of classifying insect families by a random combination of four biometric parameters (i.e., wingbeat frequency, body length, body width, and body mass).

Wingbeat Frequency	Body Length	Body Width	Body Mass	Identification Accuracy
〇	〇	〇	〇	0.845356
〇	〇	〇		0.819244
〇		〇	〇	0.763059
〇	〇		〇	0.815071
〇	〇			0.708454
〇		〇		0.703767
〇			〇	0.710949
〇				0.435523
	〇			0.456125
		〇		0.485464
			〇	0.442205

Note: “〇” represent parameters selected in the classification methods.

## Data Availability

The data presented in this study are available from the corresponding author upon request. The data are not publicly available due to privacy restrictions.

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
