# Peer review of "Characterization of Wingbeat Frequency of Different Taxa of Migratory Insects in Northeast Asia"

_insects, 2022, doi:10.3390/insects13060520_

Round 1
Author Response
We are thankful to the reviewer 1 for your comments and suggestions on our MS. We have accepted your comments and revised the manuscript accordingly. Please see the attachment.

Reviewer 2 Report
The study investigated wingbeat frequency of 3 insect orders, 15 families that arrived on a small island in Bohai Sea, eastern China. The wingbeat frequency is an important parameter observed with an entomological radar to suggest insect species. The study is characterized by abundance of insect species and number tested and analysis of determining factors of the wingbeat frequency. The results showed that the wingbeat frequency of lepidopteran insects was linearly correlated with the product of square root of mass and the inverse of wing area. This result corresponded to a previously reported Deakin’s formula in reference 26. Only weak linearity was found in Neuroptera tested and no such linearity was found in Odonata. These are quite interesting.
The methods are fine. The results support the conclusion.
I just suggest a following correction. In the last part of Discussion, "Given the difference in biological variables and ecological niches of pests and natural enemies, follow-up research eventually combined ecological niches models are needed help with early warning and monitoring of pests and natural enemies.” This part is very difficult to be understood in regarding the wingbeat frequency. I feel it's a disconnection in concept flow. I don’t see any connection between the wingbeat frequency and the ecology of pest and natural enemy. Please rephrase it.
Overall, presentation of the study is fine. Followings are minor corrections I noticed:
Lines 254-259: The results described in Discussion are not found in the previous sections, M&M and Results. Please add them properly.
Lines 268-271: I don't know what the sentence means. Please rewrite it.
Author Response
We are thankful to the reviewer 2 for your positive assessment on our MS. We have accepted your comments and revised the manuscript accordingly.

Reviewer 3 Report
I read the article by Yu et al. that was titled "Characterization of Wingbeat Frequency of Different Taxa of Migratory Insects in Northeast Asia." In general, the article appears to be of high quality and provides a useful insight into the correlations between wingbeat frequency and various developmental morphometrics. In the meantime, I have some concerns regarding the figures and the information regarding the figure legend.
1. I believe it to be quite justified to present number data information for figures either in the figure itself or in the figure legend. Also, statistical analysis for figures in legends makes it easier for readers.
2. It is not entirely clear why the square root of body mass was used for the correlation, and this is not addressed in the methods section or the discussion. It is important that this be justified in the text in an appropriate location.
3. What was the rationale behind including all three orders in the correlation studies shown in figure 4? Is there a justification for this? I believe that the independent correlations for the various orders will present more information.
4. Additional figures 6 and 8 provide data that is of less interest and somewhat duplicates the data that is shown in figure 4. As a result, it is possible that these figures should be moved into the supplementary figures section.
5. In addition, the authors need to be very careful when editing the manuscript, as it currently contains some typographical errors and duplications.
Author Response
We are thankful to reviewer 3 for your positive assessment on our Manuscript. We have accepted your comments and revised the manuscript accordingly. Please see the attachment.

Round 2
Reviewer 3 Report
We are grateful to the authors for considering the feedback and making the necessary changes to the manuscript. The article has been made better!